



**Water-level attenuation in broad-scale assessments of exposure to coastal flooding: a sensitivity analysis**

Athanasios T. Vafeidis[1], Mark Schuerch[2], Claudia Wolff[1], Tom Spencer[3], Jan L. Merkens[1], Jochen Hinkel[4], Daniel Lincke[4], Sally Brown[5], Robert J. Nicholls[5]

[1]*Coastal Risks and Sea-Level Rise Research Group, Department of Geography, Christian-Albrechts University Kiel, Ludewig-Meyn-Str. 14, 24098 Kiel, Germany*

[2]*Lincoln Centre for Water and Planetary Health, School of Geography, University of Lincoln, Think Tank, Lincoln, LN6 7FL, UK*

[3]*Cambridge Coastal Research Unit, Department of Geography, University of Cambridge, Downing Place, Cambridge CB2 3EN, UK*

[4]*Global Climate Forum e.V. (GCF), Neue Promenade 6, 10178 Berlin, Germany*

[5]*Faculty of Engineering and the Environment, University of Southampton, Highfield, Southampton SO17 1BJ, UK*

**Abstract**

This study explores the uncertainty introduced in global assessments of coastal flood exposure and risk when not accounting for water level attenuation due to land-surface characteristics. We implement a range of plausible water-level attenuation values for characteristic land-cover classes in the flood module of the Dynamic and Integrated Vulnerability Assessment (DIVA) modelling framework and assess the sensitivity of flood exposure and flood risk indicators to differences in attenuation rates. Results show a reduction of up to 47% in area exposure and even larger reductions in population exposure and expected flood damages when considering water level attenuation. The reductions vary by country, reflecting the differences in the physical characteristics of the floodplain as well as in the spatial distribution of people and assets in coastal regions. We find that uncertainties related to not accounting for water attenuation in global assessments of flood risk are of similar magnitude to the uncertainties related to the amount of SLR expected over the 21st century. Despite using simplified assumptions to account for the process of water level attenuation, which depends on numerous factors and their complex interactions, our results strongly suggest that an improved understanding and representation of the temporal and spatial variation of water levels across floodplains is essential for future impact modelling.

**1. Introduction**

Increased flooding due to sea-level rise (SLR) is a major natural hazard that coastal regions will face in the 21st century, with potentially high socio-economic impacts (Kron, 2013; Wong et al., 2014). Broad-scale (i.e. continental to global) assessments of coastal flood exposure and risk are therefore required to inform mitigation targets and adaptation decisions (Ward et al., 2013a), related financial needs and loss and damage estimates. Towards these ends, a number of recent studies have assessed the





exposure of area, population and assets to coastal flooding at national to global scales (Nicholls, 2004; Brown et al. 2013; Jongman et al., 2012a; Ward et al., 2013b; Arkema et al., 2013; Muis et al., 2017) as well as flood risk (Hinkel et al. 2014; Vousdoukas et al., 2018a).

Although methods for broad-scale coastal-flood exposure and risk assessment vary between studies, flood extent and water depth have commonly been assessed based on spatial analysis, assuming that all areas with an elevation below a certain water level that are hydrologically connected to the sea are flooded (the "bathtub" method) (Poulter and Halpin, 2008; Lichter et al., 2010). Notable exceptions are the studies of Dasgupta et al. (2009), who used a simple approach to account for wave height attenuation with distance from the coast, and Vousdoukas et al. (2018b) who, for the Iberian peninsula, adopted a modified version of the bathtub approach that also considers water volume. The use of simplified methods for assessing flooding is primarily related to difficulties of using hydrodynamic methods at broad scale, namely the limited availability and large volume of the necessary high-resolution input data; and the prohibitive computational costs, which render hydrodynamic modelling applications impractical at global scales (Ramirez et al., 2016). Therefore, global applications have utilised elevation data with a spatial resolution of 1 km and a vertical resolution of 1m (Mondal and Tatem, 2012; Jongman et al., 2012b; Ward et al., 2014), with only a few recent studies employing higher spatial resolution (90m) datasets (e.g. Hinkel et al., 2014; Vousdoukas et al., 2018a; see also de Moel et al., 2015).

Hydrodynamic models are normally used only for local-scale applications. This is because they require detailed data on parameters such as coastal topography/bathymetry and land use in order to represent local-scale processes and to account for hydraulic properties. A range of simpler inundation models that partly account for hydraulic processes at intermediate scales using medium resolution elevation data ($<100m^2$) have also been applied at subnational scales (e.g., Bates et al., 2010; Wadey et al., 2012; Lewis et al. 2015; Ramirez et al., 2016), and these models are beginning to inform analysis at broader scales (e.g. Vousdoukas et al., 2016; 2018a). There is also a developing literature on hydrodynamic modelling of water level attenuation over coastal wetlands at the landscape scale (<1 km) for saltmarshes (Loder et al., 2009; Wamsley et al., 2009, 2010; Barbier et al., 2013; Smith et al., 2016) and mangrove forests (McIvor et al., 2012; Zhang et al., 2012; Liu et al., 2013). However, the incorporation of the above processes in global models is still very limited.

Not accounting for hydrodynamic processes in global models can, however, lead to overestimation of flood extent and water depth. Hydrodynamic models capture processes that are not included in global models, e.g. the effects of surface roughness (both natural and anthropogenic) and channel network density and connectivity (and its effect on landscape continuity) on the timing, duration and routing of floodwaters. For example, inundation extent has been shown in some cases to significantly decrease



in urban and residential areas when the built environment is represented in numerical simulations (e.g.
tsunami inundation: Kaiser et al., 2011; storm surge inundation: Brown et al., 2007; Orton et al., 2015).
To our knowledge, there is no study that has explored the uncertainty introduced into global models
by not accounting for water level attenuation due to hydrodynamic processes related to surface
roughness. This paper aims to address this gap. We derive a range of plausible water-level attenuation
values from existing literature and implement them in the flood module of the Dynamic Interactive
Vulnerability Assessment (DIVA) modelling framework (Hinkel et al., 2014). Next, we assess the
sensitivity of flood exposure and flood risk indicators to plausible changes in water-level attenuation
values under a range of different SLR scenarios. Finally, we compare the uncertainty due to water level
attenuation rates with the uncertainty range associated with expected SLR during the 21st Century.

## 2. Methods and Data


*2.1 The Dynamic Interactive Vulnerability Assessment (DIVA) modelling framework*

DIVA is an integrated, global modelling framework for assessing the biophysical and socio-economic
consequences of SLR, and associated extreme water levels, under different physical and socio-
economic scenarios and considering various adaptation strategies (Hinkel and Klein, 2009). DIVA has
been widely used for global and continental scale assessments of SLR impacts, vulnerability and
adaptation (e.g., McLeod et al., 2010; Hinkel et al. 2010; Brown et al. 2013; Hinkel et al., 2013; Hinkel
et al., 2014; Spencer et al., 2016; Schuerch et al., 2018). It is underpinned by a global coastal database
which divides the world's coastline (excluding Antarctica) into 12,148 coastal segments (Vafeidis et al.,
2008). Each segment contains approximately 100 elements of data concerning the physical, ecological
and socio-economic characteristics of the coast. Here we focus on the impacts of increased exposure
to coastal flooding and potential damages of extreme sea level events (due to the combination of
storm surges and astronomical high tides). We used the flood module of DIVA (for details see Hinkel
et al., 2014) to estimate potential coastal flood damage, SLR impacts and associated costs.
We specifically considered the following five indicators, which progressively include additional
components of flood risk:

102       1. Area below the 1-in-100 year flood event ($km^2$), an estimate based on elevation data and
103          information on water levels for a single hazard event (i.e. the height of the 1-in-100 year sea
104          flood);



2.  People living in the 1-in-100 year floodplain, a calculation based on spatial data on elevation
and population as well as on information for a single hazard event (i.e. the height of the 1-in-
100 year sea flood);
3.  Assets in the 1-in-100 year floodplain (US $), a calculation that uses data on elevation,
population, Gross Domestic Product (GDP) and information for a single hazard event (i.e. the
height of the 1-in-100 year sea flood);
4.  Expected value of the number of people flooded per year (hereafter, people flooded), a
calculation based on elevation and population data and the probability distribution of the
hazard (i.e. sea flood heights and their probability of occurrence); and
5.  Expected value of annual damages to assets (hereafter, flood damage) (US $), a calculation
based on elevation, population and GDP data and the probability distribution of the hazard
(i.e. sea flood heights and their probability of occurrence).
For each coastline segment, a cumulative exposure function for area and population that gives the
areal extent (hydrologically connected to the sea) and number of people below a given elevation was
constructed. Damages to assets were assessed using a depth-damage function with a declining slope,
with 50% of the assets being destroyed at a water depth of one metre (Messner et al., 2007).

*2.2 Coastal Elevation and Rate of Water level Attenuation*
To simulate the effect of different values of attenuation at the broad scale, we implemented a stylised
elevation profile to represent the process of water level attenuation. We assumed that water levels
decrease at a constant slope ($\alpha$) with increasing distance from the coastline. Location-specific coastal
profiles for every coastline segment were based on floodplain areas contained within the DIVA
database. The database reports total land area within different elevation increments (<1.5m, 1.5-2.5m,
2.5-3.5m, 3.5-4.5m, 4.5-5.5m, 5.5-8.5m, 8.5-12.5m, 12.5-16.5m) for each coastal segment. The
elevation dataset that was used for estimating floodplain areas and developing the segment elevation
profiles is the Shuttle Radar Terrain Mission (SRTM) Digital Elevation Database (Jarvis et al., 2008)
which has a vertical resolution of 1m and a spatial resolution of 3 arc seconds (~90m at the equator).
We approximated the average coastal profile for every segment by assuming that elevation
continuously increases with distance from the shore. Starting with the lowest elevation increment, the
floodplain areas of all elevation increments were cumulatively summed to retrieve the total area below
a certain elevation. The total areas were then divided by the segment length to derive the inundation
length of the respective floodplain ($dx_i$). To evaluate the representativeness of the assumption of
continuously increasing elevation with increasing distance from the shore, we used the original SRTM
dataset and calculated the Euclidian distance of each cell to the nearest coastline for every pixel. Mean
distances from the coast were calculated for each of the floodplain areas of each segment.
Subsequently, we compared these mean distances with the respective average floodplain elevation





for each DIVA coastline segment to analyse the validity of the "continuous-increase" assumption. This
comparison revealed that 55% of the DIVA coastline segments show either a continuous increase or
no change in the mean distance along the elevation profile (Figure 1a), suggesting that elevation does
not decrease with distance from the coast. Comparing all elevation increments of all segments (i.e.
pairwise comparison of the mean distances of consecutive elevation increments in a segment), there
was an increase, or no change, in the mean distance from the coastline in 88% of cases. Only 12% of
cases showed a decrease (Figure 1b). This result indicates that the stylised continuous profile (Figure
1a) can be regarded as representative of global coastal topography (see also Schuerch et al., 2018).



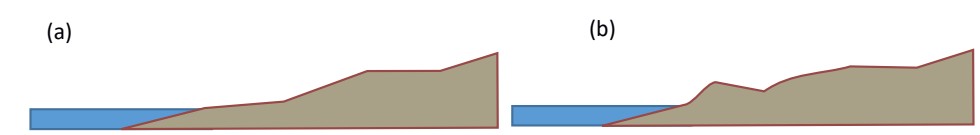

**Figure 1:** Stylised coastal profile with (a) continuous and (b) discontinuous increase in elevation with distance
from the shore.
We then adjusted the coastal profile using a range of possible attenuation rates that represent
different water surface slopes. Depending on the applied value for water level attenuation, the slope
($\alpha$) of the inundating water surface was employed to modify (incline) the coastal profile. Based on this
slope, the coastal profile is thereby elevated by the amount of the water level reduction ($hx_i$) computed
at a distance $dx_i$ (Fig. 1):
$$hx_i = tan(\alpha) * dx_i \qquad \text{(equation 1)}$$
In this way the original floodplain areas and inundation depths are reduced in order to account for the
reduced (i) inundation length ($dx$) and (ii) inundation depth ($hx$) (see Fig. 2).



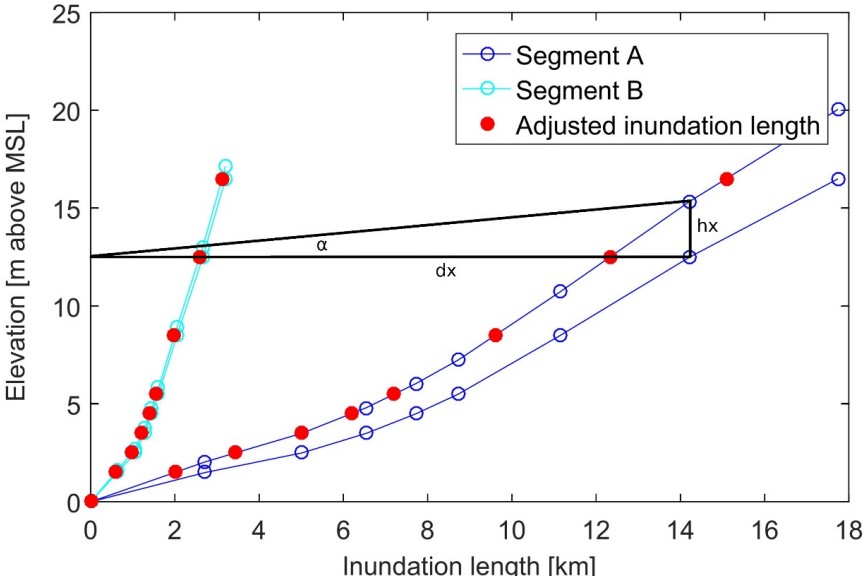


**Figure 2**: The stylised coastal profile, based on the floodplain areas in the DIVA database (lower line), for two
characteristic coastline segments (A with a flat and B with a steep profile). Water level attenuation is accounted
for by inclining the coastal profile according to equation 1 (upper line). Red dots on the adjusted coastal profile
indicate the inundation length in the case of a water level with a constant slope of $\alpha$, which represents the
attenuation rate and for an incident water level equal to the corresponding increment height.


For the sensitivity analysis we used a range of attenuation rates that embraces the values reported in
the literature (Table 1), where water level under storm conditions has been shown to decrease with
distance from the coast. For reviewing the literature we employed the ISI Web of Knowledge and based
our search on the keywords "surge", "attenuation", "water-level". We selected studies that directly
reported values of water level reduction with distance and did not include studies focussing on wave
attenuation. We must note that the aim was not to conduct a systematic literature review but rather
to identify a characteristic range of values that could support the sensitivity analysis. The identified
studies all relate to coastal wetland environments. Although there are published studies of localised
water level dynamics from flow-form interactions in urban and other settings, we have not come
across similar landscape-scale assessments for other land use types. Therefore we broadened this
review, where reported attenuation values were up to 70cm/km, by directly contacting scientists and
data analysts with experience in field or modelling studies. Following their expert advice , we extended
our analysis to include attenuation rates of up to 100 cm/km as an upper limit.



| Event type | Landcover type | Location | Rate of water-level reduction | Method | Source |
|---|---|---|---|---|---|
| Storm surge | Bare land and Marsh | Modelled platform +0.5 m above sea level | 10 cm / km (no vegetation, no channels) 26 cm / km (100% vegetation cover, no channels) 8 cm / km (100% vegetation cover, channel network) | Numerical modelling | Temmerman et al., 2012 |
| Hurricane Isaac (2012) | Marsh | Louisiana | Up to 70cm/km water level reduction in presence of vegetation; 37 % reduction of total inundation volume | Numerical modelling | Hu et al., 2015 |
| Hurricanes | Marsh | Multiple | 1 m per 14.5 km 6.9 cm/km (range from 1m per 5km to 1m per 60km 20 - 1.7 cm/km) | Field Study | Corps of Engineers (1963) – In Wamsley et al., 2010 |
| Hurricane Andrew (1992) | Marsh | Louisiana | 1m per 20km-23.5km 5 - 4.3 cm/km | Field Study | Lovelace 1994 |
| Hurricane Rita (2005) | | Louisiana | 1m per 4km to 1m per 25km 25 – 4 cm/km | Field Study | McGee et al. 2006 in Wamsley et al., 2010 |
| Hurricanes Wilma (2005) and Charley (2004) | Mangroves Marsh | Florida | 9.4 - 4.2 -cm/km | Field Study | Krauss et al., 2009 |
| Hurricanes | Mangroves | Louisiana | 23.3 – 1.7 cm/km | Field Studies | McIvor et al., 2012 (from various studies) |
| Hurricane Wilma (2005) | Mangroves | South Florida | Up to 50 cm/km (6-10 cm per km in the absence of mangroves) | Field study & modelling | Zhang et al., 2012 |
| Hurricanes | Mangroves | South Florida | 7.7 - 5.0 cm/km | Modelling | Liu et al., 2013 |

**Table 1**: Water level reduction rates, for different types of landcover, as reported in the literature.

We further constrained the sensitivity analysis by adjusting the range water attenuation rate for each
segment based on the predominant land use type covering the area of every elevation increment. For
estimating the predominant land use we employed the GlobCover Land Cover V2.3 dataset, a global
land cover dataset with a resolution of 10 arc second (~300 meter at the equator). It is based on
the ENVISAT satellite mission's MERIS sensor (Medium Resolution Image Spectrometer) covering the
period between January and December 2009 and includes 22 land cover classes. As the available
information on water attenuation rates by land use type is limited, we reclassified the data to seven





classes (forest, urban, cropland, grassland, mangroves, saltmarshes and Unknown) and assigned
maximum attenuation rates to each class (Table 2). For the model runs we used the five attenuation
categories (no, low, medium, high and maximum attenuation) corresponding to 0, 25%, 50%, 75% and
100% of the maximum values found in the literature / from expert review, for each class. These rates
were then used to incline the water surface in order to represent a constant water level attenuation
and the associated reduction in water levels ($\alpha$) across the floodplain for each coastline segment.

| Land Use Class | Maximum Attenuation (cm/km) |
|---|---|
| FOREST (1) | 50 |
| URBAN (2) | 100 |
| CROPLAND (3) | 40 |
| GRASSLAND (4) | 25 |
| MANGROVES (5) | 50 |
| SALTMARSHES (6) | 25 |
| UNKOWN (0) | 25 |

Table 2: Maximum attenuation rates per land use class used in the sensitivity analysis
*2.4 Sea-Level Rise and Socio-Economic Scenarios*
For global SLR in 2100 from a 1985 – 2005 baseline we used three scenarios: the 5% quantile of the
low Representative Concentration Pathway (RCP) 2.6; the median of the medium scenario RCP 4.5;
and the 95% quartile of the high scenario RCP 8.5. These scenarios are represented by SLR estimates
of 29, 50 and 110 cm (by 2100), respectively and were developed in the Inter Sectoral Model
Intercomparison Project Fast Track (for full details see Hinkel et al., 2014). Following Menendez and
Woodworth (2010), once mean sea level had been determined, future extreme water levels were
obtained by displacing upwards extreme water levels for different return periods (as included in the
DIVA database) with the rising sea level.
We used a single shared socio-economic pathway (SSP), namely SSP2, to represent changes in coastal
population and assets. SSP2 reflects a world with medium assumptions between the other four SSPs,
in terms of resource intensity and fuel dependency as well as GDP and population development (O'Neil
et al., 2014). Finally, we ran the DIVA model using a no-dike scenario, where no defence measures for
preventing coastal flooding are present.

**3. Results**
We present results for the different classes of attenuation rates, across the five indicators that
progressively include additional components of flood risk:

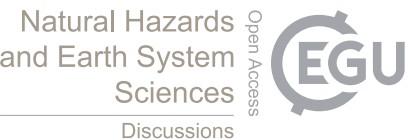


### 3.1 Reduction of current flood exposure and risk

Table 3 shows the results from the five categories of attenuation rates and both the absolute and percentage reductions in the values of the five indicators against this baseline.

| | Water Level Attenuation Category | | | | |
|---|---|---|---|---|---|
| | NO | LOW (% decrease) | MEDIUM (% decrease) | HIGH (% decrease) | FULL (% decrease) |
| Area below the 1-in-100 year flood [km²] | 859,059 | 654,474 (24%) | 572,546 (33%) | 519,871 (40%) | 481,379 (44%) |
| People below the 1-in-100 year flood [million] | 196 | 133 (32%) | 115 (41%) | 104 (47%) | 97 (51%) |
| Assets below the 1-in-100 year flood [billion US$] | 52,663 | 34,814 (34%) | 29,486 (44%) | 26,467 (50%) | 24,567 (53%) |
| People flooded [million/yr] | 4429 | 3086 (30%) | 2673 (40%) | 2341 (47%) | 2156 (51%) |
| Flood damages to assets for the 1-in-100 year flood [billion US$/yr] | 3,577 | 2,361 (34%) | 1,972 (45%) | 1,782 (50%) | 1,660 (54%) |

**Table 3**: Reduction, relative to the bathtub method, of five indicators of global exposure and risk, for different water-level attenuation rates. Values are for a medium SLR scenario (median of the medium scenario RCP 4.5; 50 cm by 2100)

Our results show that accounting for water-level attenuation in the assessment of flooding results in large differences in the values of the five indicators. For example, the area exposed to the 1 in 100-year flood in 2015 reduces by up to 44% with the application of attenuation rates. The low attenuation category results in an area reduction of 24% while the use of medium attenuation rates results in a reduction of 33% (see Table 3). Interestingly, the number of people in the 1 in 100-year floodplain reduces to 97 million when considering high attenuation. This is a reduction of 51%, which is similar to the respective reduction in assets (53%) but higher than the reduction in area (44%) exposure. This result reflects the high population density near the coast that has been reported in previous studies (e.g. Neumann et al., 2015). Flood damages from the 1-in-100 year event are reduced in similar proportion, totalling a reduction of more than 1.9 trillion US$ (54%) globally, when considering maximum attenuation rates.

The reduction in impacts is not uniform across the globe and varies considerably between different countries. Some examples are given in Table 4, where accounting for water level attenuation reduces area exposure by up to 73% in China, 40% in Bangladesh and 50% in the USA. At the same time, the





reduction in annual flood costs follows a different trend, with exposed assets reducing by up to 71% in
China, 42% in Bangladesh and 25% in the USA, reflecting differences in the physical characteristics of
the floodplain as well as in the spatial distribution of people and assets in the coastal regions of these
countries.

| Water Level Attenuation Rate | 0 cm/km | 10 cm/km (reduction) | 20 cm/km (reduction) | 50 cm/km (reduction) | 100 cm/km (reduction) |
|---|---|---|---|---|---|
| **Area below 1-in-100 year flood** | | | | | |
| (km$^2$) | | | | | |
| Bangladesh | 5831.4 | 4642.7 | 4204.7 | 3864.1 | 3528.3 |
| | | (20%) | (28%) | (34%) | (40%) |
| China | 85864.8 | 43770.7 | 32588.3 | 27018.2 | 23413.4 |
| | | (49%) | (63%) | (69%) | (73%) |
| USA | 69924.1 | 54244.5 | 43528.3 | 39346.5 | 35386.5 |
| | | (24%) | (39%) | (44%) | (50%) |
| | | | | | |
| **Assets below 1-in-100 year flood** | | | | | |
| (billion $US) | | | | | |
| Bangladesh | 67.2 | 54.7 | 48.8 | 43.6 | 39.4 |
| | | (19%) | (27%) | (35%) | (42%) |
| China | 5137.8 | 2319.6 | 1724.2 | 1432.4 | 1258.1 |
| | | (55%) | (66%) | (72%) | (76%) |
| USA | 533.7 | 431.4 | 388.2 | 360.7 | 341.9 |
| | | (19%) | (27%) | (33%) | (36%) |


**Table 4**: Absolute and relative reduction of the 1-in-100-year floodplain area and associated exposed assets when
applying different water-level attenuation rates for Bangladesh, China and USA. Values assume a medium SLR
scenario (median of the medium scenario RCP 4.5; 50 cm in 2100).

*3.2 Comparison of attenuation rate uncertainty with sea-level rise uncertainty*

Figure 3 illustrates the area of land located below the 1-in-100 year storm surge level (H100), plotted
against the different attenuation rates for water level change. The inclusion of water-level attenuation
in the assessment of flooding results in large reduction in the extent of the 100-year floodplain in 2100
(Figure 3) under all SLR scenarios. Even the use of low attenuation of water levels results in a reduction
of 230,000 km$^2$ of area exposed to the 1-in-100-year flood under the no-SLR scenario. This increases
to 350,000 km$^2$ under the high SLR scenario. For the medium SLR scenario (median of the medium


scenario RCP 4.5; 50 cm by 2100), this reduction amounts to 31% and 40% of the total exposed area
at medium and full water level attenuation respectively. The relative reduction is larger (up to 60%)
for the high SLR scenario compared to the medium-, low- and no-SLR scenarios. Importantly, the
overall difference in the extent of the area of the 100-year floodplain between the no- and high-SLR
scenarios is of a similar order of magnitude to the difference in area extent between the no and low
water level attenuation rates, under any scenario. This indicates that when assessing area exposure
accounting for even relatively moderate rates of water level attenuation can be of similar importance
to the differences that result from different scenarios of SLR. This analysis, therefore, strongly suggests
that uncertainties related to the omission of this factor in global assessments of flood risk are of similar
magnitude to the uncertainties related to the magnitude of SLR expected over the 21st century.

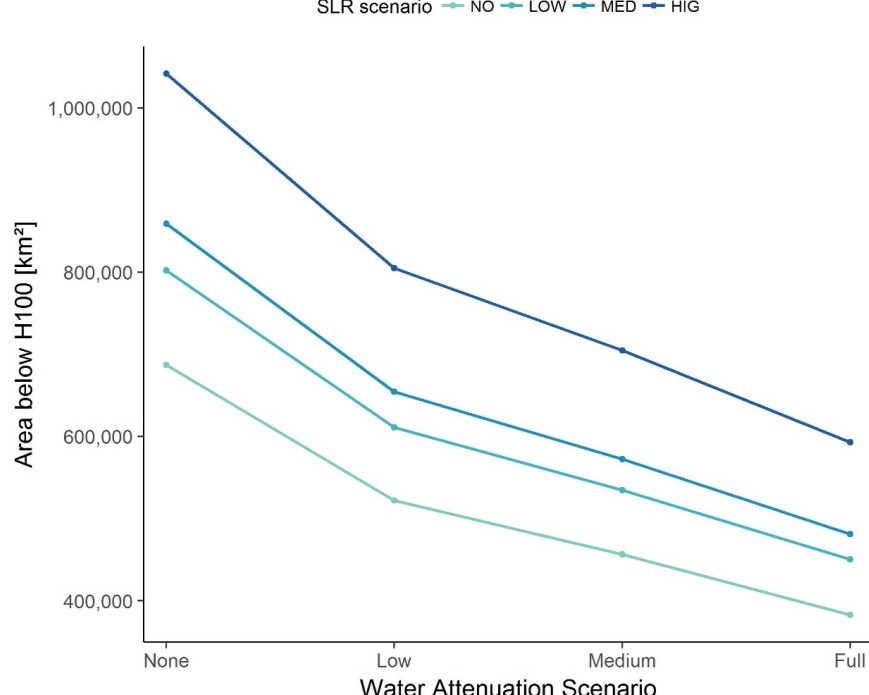


**Figure 3:** Global total extent of the 100-year floodplain, for different water level attenuation rates and SLR
scenarios.

Similar patterns can be observed for the exposure of population to the 1-in-100-year flood (Figure 4).
Low attenuation (Table 1), leads to a reduction of more than 30% in the exposure of population in
2100, under the high SLR scenario, bringing the number of people at risk in the 100-year floodplain





down by approximately 75 million. Moreover, medium attenuation leads to a reduction in flood
exposure by 100 million people, making population exposure lower than the exposure under no SLR
when attenuation is not considered. Again, this result suggests that accounting for water level
attenuation may be equally important to accounting for SLR uncertainty when assessing the exposure
of people to coastal flooding due to SLR.

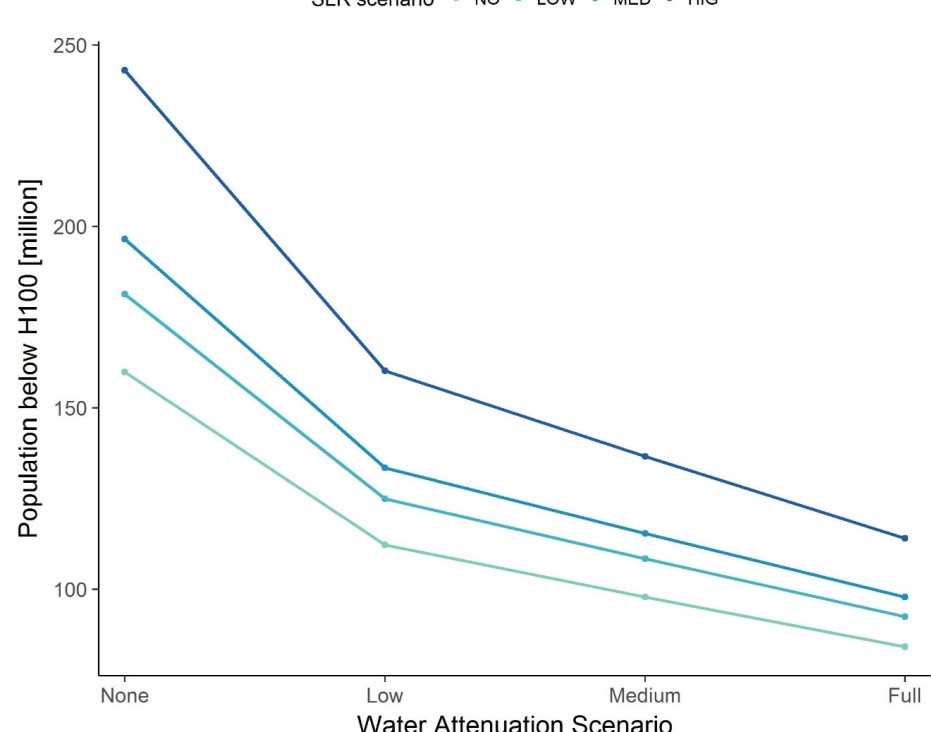


**Figure 4**: Global estimates of population in the 100-year floodplain for different water-level reduction rates
(Table 1) and SLR scenarios.
The value of assets exposed to the 1-in-100-year flood, under all scenarios, are also reduced
substantially when accounting for water level attenuation (Figure 5). Considering low attenuation rates
results in a decrease in the exposure of assets of approximately 34% in 2100, for a medium SLR
scenario. A reduction of 50% in assets' exposure occur when high attenuation is used. Furthermore,
the use of a relatively moderate attenuation rate shifts the impact on assets' exposure by
approximately 30 years, under all SLR scenarios.





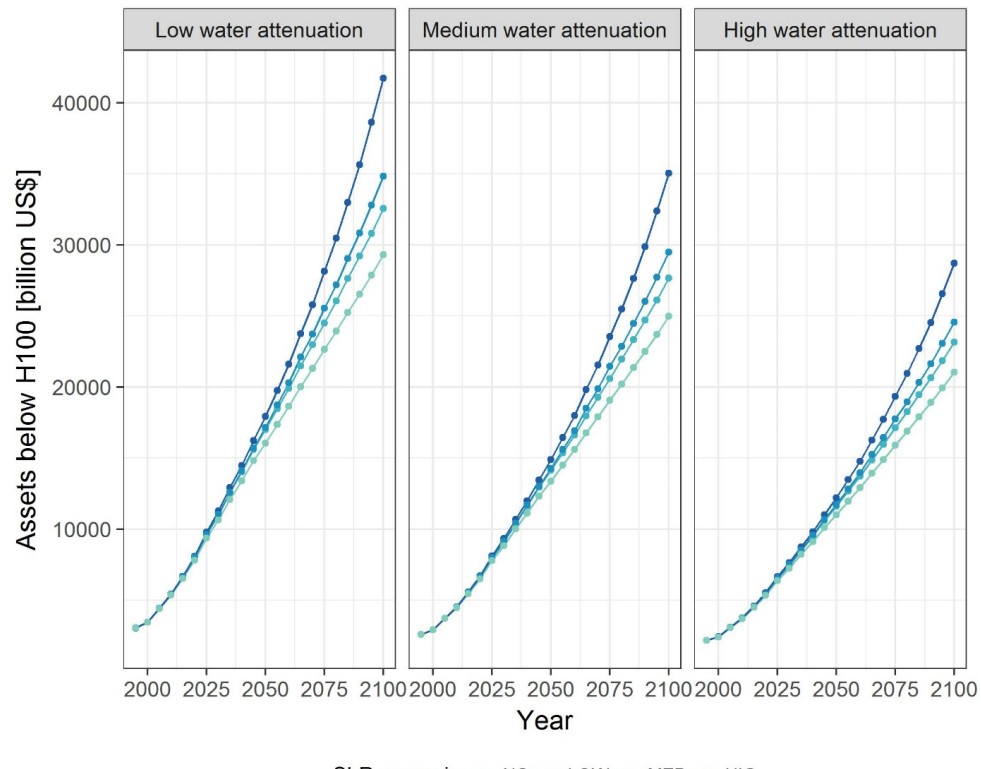


**Figure 5**: Temporal evolution of the amount of assets that are located in the 100-year floodplain for different water-level reduction rates (Table 1) and SLR scenarios.




Damages also reduce considerably with the introduction of water level attenuation rates (Figure 6).
For example, the use of a low attenuation rate results in a 34% reduction in damages to assets in 2100
from the 1-in-100 year flood. The larger decrease in damages due to water level attenuation compared
to population and area exposure is due to the fact that, besides the decrease of the flood area extent,
water level attenuation leads to an additional reduction of flood depth with distance from the coast.
As water depth is an important parameter for calculating damages to assets (Thieken et al., 2005;
Penning-Rowsell et al., 2013), depth reduction further reduces the potential damages of assets due to
flooding.





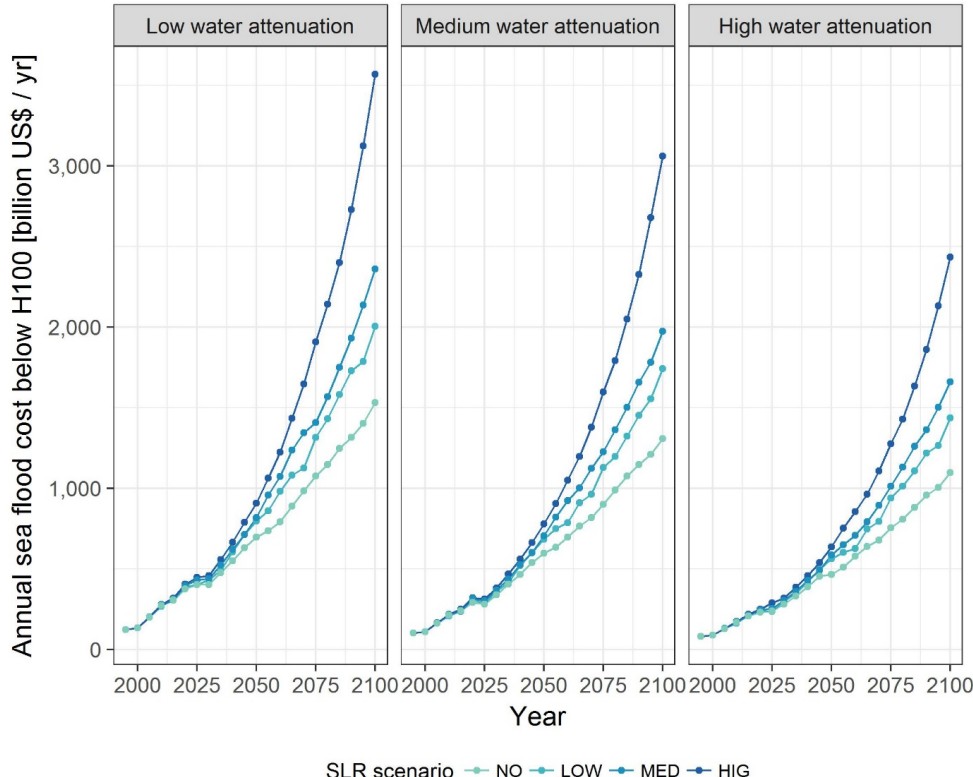


Figure 6: Comparison of temporal evolution of sea-flood damage estimates for attenuation rates of 0,
10 and 50 cm/km, for different SLR scenarios.

**4. Discussion and Conclusions**
This study highlights the importance of accounting for the effects of hydrodynamic processes when
assessing the impacts of coastal flooding at national to global scales. In particular, water level
attenuation from the interaction of extreme inundation events with vegetated surfaces can lead to
considerably lower estimates of exposure of land area and population to coastal flooding.
Furthermore, this effect can lead to large reductions in potential damages, as lower water depths
combined with smaller flood extents give significantly lower flood-damage costs. The reduction in
exposure and risk is very pronounced, even when considering low water level attenuation rates.
Accounting for water level attenuation appears to be as important in assessing impacts as accounting
for uncertainties related to the total magnitude of SLR. In many of the cases explored, the difference
in impacts between no- and high-SLR scenarios is similar to the difference in impacts between no- and



low attenuation rates of up to 12.5 cm/km (excluding urban land use). This finding is of particular
relevance in environments where the floodplain substantially extends inland, such as in many of the
world's deltas and coastal plains.
It is widely acknowledged that the use of simplified methods, such as the bathtub method, can provide
useful first-order estimates of global impacts of SLR and associated flooding (Lichter et al., 2010; Hinkel
et al., 2014), although an overestimation of flood extent and depth with the use of the bathtub method
should be generally anticipated (Vousdoukas et al., 2016). Further, we must note that the reduction
that we observe with the use of water level attenuation rates does not necessarily reflect actual
impacts. These are likely to depend on additional factors which are usually not considered in global
assessments. For example, damage to assets in our analysis is based solely on water depth; factors
such as high local flow velocities from channelized flow, storm wave impacts, inundation by saline
water and sedimentation from flood waters are not taken into account. Such contributory factors can
lead to an increased cost of damages and thus counteract the lower impacts predicted from the use of
a water level attenuation term alone. Furthermore, the analysis reported here is predicated on the
assumption of a continuous increase in elevation with increasing distance from the shore. This study
shows that whilst this assumption is valid for the majority of coastal segments, there are segments
where this assumption does not hold true. In these cases model outputs may poorly describe flood
areas, flooded population numbers and asset damages and incorrectly predict the effect of changes in
the rate of water level attenuation. Nevertheless, and despite these caveats, our results emphasise
the importance of accounting for uncertainties in impact assessments stemming from the lack of
consideration of water level attenuation over coastal plains.
Our approach means to provide an illustration of the potential effects of water level attenuation, as
this process is not constant throughout the floodplain and depends on numerous parameters beyond
the type of the surface cover. These factors include storm duration, wind direction, water depth and
vegetation traits (Resio and Westerink, 2008; Smith et al., 2016; Stark et al., 2016). Furthermore,
applying a constant slope to account for water level attenuation is a strong simplification, since this
will vary between different storm events, but also under the influence of SLR. Nevertheless, given the
very high sensitivity of the outputs to even small changes in water level reduction rates; and the
obvious lack of sufficient data on the actual effect of different types of surface on attenuating water
levels during surges, we suggest that future work needs to focus on quantifying the water level
attenuation terms for different land uses. Thus, for example, both Brown et al. (2007), in the case of
modelled flooding following storm surge-induced sea defence failure, and Kaiser et al. (2011), in the
case of modelled tsunami wave impacts, have shown that disregarding buildings and associated
infrastructure (roads, gardens, ditches) when assessing inundation can lead to a large overestimation

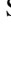



of the extent of flooding. Furthermore, given the large range of uncertainty with respect to the actual
values of water level reduction associated with just one surface cover, wetland habitat (Table 1), future
impact modelling needs to focus on a better understanding of the temporal and spatial variation of
water levels across floodplains that show a wide variety of land use types and human occupancy,
including densely urbanised regions (e.g. Lewis et al., 2013; Blumberg et al., 2015).
Given that coastal wetlands can efficiently attenuate surge water levels, the results of this study give
a first estimate of how much of an impact reduction may result from the implementation of large-
scale, ecosystem-based flood risk reduction management schemes (e.g. Temmerman et al., 2013). In
addition, achieving lower water levels through the establishment of coastal wetlands not only reduces
impacts but may also affect the timing of potential adaptation tipping points by extending the
anticipated lifetime of adaptation measures. This would allow the development of alternative
adaptation pathways, a sequential series of linked adaptation options triggered by changes in external
conditions (Barbier, 2015), for coastal regions.

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
