# Peer review of "Water-level attenuation in global-scale assessments of exposure to coastal flooding: a"

_Natural Hazards and Earth System Sciences, 2018_

## Referee Comment (RC1) · Anonymous Referee #1 · 11 Feb 2019

Review of Vafeidis et al. 2019, NHESSD

This paper proposes a new scheme implemented into the framework of coastal flood risk assessment to consider water level attenuation. I think the paper gives new and important insights that the effect of water level attenuation is significantly large, which in some cases are equal to uncertainty of sea level rise projection. The topic of this paper fits well with NHESS and the manuscript is well written and easy to follow. Therefore, I recommend this paper for publication in NHESS after some minor revisions, which are listed below.

P 1, L 1 Basically the target domain of this paper is global, so I think the "broad-scale"

[Figure]

in this title should be reworded as "global-scale" for clarity.

P 4, L 137–138 It is well known that the original SRTM DEM has strong biases. In particular, elevation data in coastal areas are strongly biased by vegetation, e.g. mangrove forest in Bangladesh. This factor may significantly affect the results of global coastal flood risk estimation. Recently a new DEM dataset has been developed where various types of biases are removed (Yamazaki et al. 2017). I do not request the authors to update the input topography data and redo all the analyses, but at least this limitation should be discussed in the manuscript.

P 9, Table 3 Better to replace "People" as "Population".

P 10, L 244–246 I think additional explanations are required about how the three countries are different in terms of "the physical characteristics of the floodplain".

Reference Yamazaki, D., D. Ikeshima, R. Tawatari, T. Yamaguchi, F. O'Loughlin, J. C. Neal, C. C. Sampson, S. Kanae, and P. D. Bates (2017), A high‐accuracy map of global terrain elevations, Geophys. Res. Lett., 44, 5844–5853, doi:10.1002/2017GL072874.

---

## Referee Comment (RC2) · Anonymous Referee #2 · 11 Feb 2019

The manuscript presents results from the first global analysis on the effect of including water level attenuation into broad-scale coastal impact models. The key message is a very important one, namely that uncertainties from ignoring water level attenuation can be as large as from the expected sea level rise itself by the end of the century. The paper is very well written and structured and I enjoyed reading it. I feel that the authors are honest about limitations that are inherently associated with the type of global model that is used here. But even though one can express doubts regarding the absolute numbers presented, the conclusions drawn from the analysis are sound and, as mentioned above, very important to get across to the scientific community. I have no objection against the paper being published with NHESS after the minor comments

below have been addressed.

l. 204 Is this constant sea level rise? If so it should be explicitly mentioned and briefly explained why regional projections weren't used instead. I think for this type of sensitivity analysis it might actually be better to use constant values because otherwise it would be hard to disentangle what drives higher/lower attenuation rates in different regions. Nevertheless, I think it's worth mentioning.

l. 206 I think it should be "quantile"

l. 215 What is the rational of running the no-dike scenario? Is there one?

Tab. 4 I was wondering if an additional map that shows country-by-country results (maybe only for a medium attenuation rate) would be nice to really showcase the spatial variability. The three selected countries can still be highlighted in the table.

Fig. 5. I was a bit puzzled by the change in formatting between Figs. 3&4 and Figs. 5&6. Why is the temporal evolution (more) relevant for assets and flood damage compared to population and area in the flood plain?

l. 294 Why the switch to "reduction rates"?

---

## Author Comment (AC1) · 2 Apr 2019

We would like to thank the reviewers for their useful comments and for taking the time to review the manuscript. We have addressed all the comments of the reviewers and we believe that the manuscript has improved. Our responses to each comment can be found below. The original comments of the Reviewers are also included. Please also note that we have made some additional corrections to table 3 and rounded up some figures in the text – these changes do not affect in any way the results and conclusions of the study.

Response to Referee #1

[Figure]

- This paper proposes a new scheme implemented into the framework of coastal flood risk assessment to consider water level attenuation. I think the paper gives new and important insights that the effect of water level attenuation is significantly large, which in some cases are equal to uncertainty of sea level rise projection. The topic of this paper fits well with NHESS and the manuscript is well written and easy to follow. Therefore, I recommend this paper for publication in NHESS after some minor revisions, which are listed below.

We would like to thank the reviewer for the positive comments. Below we address the points that the reviewer has raised.

- P 1, L 1 Basically the target domain of this paper is global, so I think the "broad-scale" in this title should be reworded as "global-scale" for clarity.

We agree with the reviewer and have revised the title as suggested

- P 4, L 137–138 It is well known that the original SRTM DEM has strong biases. In particular, elevation data in coastal areas are strongly biased by vegetation, e.g. mangrove forest in Bangladesh. This factor may significantly affect the results of global coastal flood risk estimation. Recently a new DEM dataset has been developed where various types of biases are removed (Yamazaki et al. 2017). I do not request the authors to update the input topography data and redo all the analyses, but at least this limitation should be discussed in the manuscript.

Thank you for the comment. The reviewer is indeed right that an updated DEM has been developed and this is now acknowledged in the manuscript. The reason why we employed the original SRTM DEM was that it was the most commonly employed in past studies. We have now added the suggested reference and some text (lines 349-352) discussing the new DEM and potential implications for this type of work. The text now reads: "New improved versions of the SRTM elevation model (Yamazaki et al., 2017) may help to partly address this limitation, while the lack of open access elevation data of higher accuracy and resolution still constitutes a significant limitation for global

studies (Schumann and Bates, 2018)."

- P 9, Table 3 Better to replace "People" as "Population".

Revised to "number of people" as we believe that this formulation is clearer

- P 10, L 244–246 I think additional explanations are required about how the three countries are different in terms of "the physical characteristics of the floodplain".

Thank you for the comment. We agree that "physical characteristics" was not well defined. We have reworded to clarify this point (lines 257-258).

Response to Referee #2

- The manuscript presents results from the first global analysis on the effect of including water level attenuation into broad-scale coastal impact models. The key message is a very important one, namely that uncertainties from ignoring water level attenuation can be as large as from the expected sea level rise itself by the end of the century. The paper is very well written and structured and I enjoyed reading it. I feel that the authors are honest about limitations that are inherently associated with the type of global model that is used here. But even though one can express doubts regarding the absolute numbers presented, the conclusions drawn from the analysis are sound and, as mentioned above, very important to get across to the scientific community. I have no objection against the paper being published with NHESS after the minor comments C1 NHESSD Interactive comment Printer-friendly version Discussion paper below have been addressed.

We would like to thank the reviewer for the positive and constructive comments. Below we have addressed the points the reviewer has raised.

- l. 204 Is this constant sea level rise? If so it should be explicitly mentioned and briefly explained why regional projections weren't used instead. I think for this type of sensitivity analysis it might actually be better to use constant values because otherwise it would be hard to disentangle what drives higher/lower attenuation rates in different

regions. Nevertheless, I think it's worth mentioning.

We have used regional projections and have now clarified this in the text (line 211). We fully agree with the reviewer that global mean values would be very useful for the sensitivity analysis; this is a very interesting comment as in most cases reviewers tend to ask for regionalised projections. However, in this analysis we opted for regionally variable projections as these are the most commonly used in previous studies.

- l. 206 I think it should be "quantile"

The reviewer is right, this was a typo. Revised as suggested.

- l. 215 What is the rational of running the no-dike scenario? Is there one?

We used a no-dike scenario because there are currently no consistent data on coastal protection at global scale; and also to reduce complexity (as dikes in DIVA are modelled). We have added some text in lines 220-223 to clarify this point. The text now reads: "Finally, we ran the DIVA model using a no-dike scenario, where no defence measures for preventing coastal flooding are present. This was done to better characterise water attenuation and to reduce complexity as dike heights in DIVA are modelled since no consistent global data on coastal protection exist (Schuerch et al., 2018)."

- Tab. 4 I was wondering if an additional map that shows country-by-country results (maybe only for a medium attenuation rate) would be nice to really showcase the spatial variability. The three selected countries can still be highlighted in the table.

We have followed the reviewer's suggestion and compiled a map with this information. This is now included in the manuscript as Figure 4. Further, we added some lines in the text (lines 246-249) to reflect the new figure. The map is attached and the text included now in the manuscript reads: "The reduction in impacts is not uniform across the globe and varies considerably between different countries. Some examples are given in Figure 4 and Table 4. Figure 4 shows the spatial variability of the effects of accounting for water attenuation: low water attenuation can lead to reductions in area

exposure of more than 50% and high attenuation can reduce area exposure by more than 80%."

- Fig. 5. I was a bit puzzled by the change in formatting between Figs. 3&4 and Figs. 5&6. Why is the temporal evolution (more) relevant for assets and flood damage compared to population and area in the flood plain?

Figures 5 & 6 aim to emphasise the temporal dimension of accounting for water attenuation and namely the temporal shift in impacts when attenuation is considered. We have emphasised this temporal dimension in the text (lines 291-292 and 304).

- l. 294 Why the switch to "reduction rates"?

We agree, this was indeed confusing. We have changed it to "attenuation rates".

[Figure]

Fig. 1.